# Antecedents of Cloud Gaming Acceptance among Gen Z: Achieving Sustainability in the Digital Gaming Industry

**Ma. Janice J. Gumasing** [1,*], **Ron Fourier B. Alonzo** [2], **Jose Mari V. Nazareno** [2] **and Ken Lance D. Guinto** [2]

1 School of Industrial Engineering and Engineering Management, Mapúa University, Manila 1002, Philippines
2 Young Innovators Research Center, Mapúa University, Manila 1002, Philippines; jmvnazareno@mymail.mapua.edu.ph (J.M.V.N.)
* Correspondence: mjjgumasing@mapua.edu.ph; Tel.: +63-(2)-8247-5000

**Abstract:** The rapid increase in mobile phone usage among Gen Z online gamers has become more prevalent due to the pandemic. However, the limited processing power of mobile devices has prompted the need for alternative solutions, such as cloud gaming. As an alternative, cloud gaming could be used to lessen the expenses and have the processing power of a high-end computer. Cloud gaming allows users to stream games on various devices, including low-end devices such as smartphones and older computers. This extends the lifecycle of hardware by enabling users to access the latest games without the need for constant hardware upgrades. As a result, e-waste generation decreases, reducing the environmental impact associated with the production and disposal of gaming hardware. This study aimed to determine the acceptance of Gen Z towards cloud gaming as a potential solution in achieving sustainability in the digital gaming industry. The extended technology acceptance model (TAM) was utilized to explore the potential for cloud gaming adoption in the Philippines and the attitudes of Gen Z towards this technology. It was found out that for the average Filipino user, attitude plays the highest significant role in cloud gaming adoption. On the other hand, perceived enjoyment and perceived ease of use were found to have a significant influence on attitude, which in turn, affects the behavioral intention to use cloud gaming among Gen Z. Thus, cloud gaming presents a promising avenue for sustainability in the gaming industry, combining energy efficiency, resource optimization, reduced e-waste, and improved accessibility.

**Keywords:** Gen Z; mobile phones; cloud gaming; technology acceptance model; behavioral intention

## 1. Introduction

This study aimed to determine the acceptance of Gen Z towards cloud gaming as a potential solution to achieve sustainability in the digital gaming industry. The extended technology acceptance model (TAM) was utilized to explore the potential for cloud gaming adoption in the Philippines and the attitudes of Gen Z towards this technology.

Online gaming is a growing industry in the Philippines, with earnings predicted to skyrocket in 2020 compared to previous years [1]. Gaming on consoles, mobile devices, and personal computers is becoming increasingly popular among Filipinos, particularly the younger population [2]. Despite an increasing preference for mobile phones for gaming due to the availability of low-cost smartphones and mobile Internet infrastructures, gaming consoles remain popular. The mobile gaming market is expected to grow over the next five years, with sales reaching USD 2.18 billion in 2027 [3].

Since the price of GPUs and other computer components continues to rise, cloud gaming is considered the industry's future [4]. The concept of cloud gaming has existed for several years; however, there is no official cloud gaming service in the Philippines. There is no concrete explanation for why the Philippines has no official cloud gaming service that does not require a virtual private network (VPN). Still, it was assumed that it might be due to the country's Internet speeds and that cloud gaming has not yet become ubiquitous [5].

Anyone in the Philippines gaming community may benefit from cloud gaming because it supports all devices if the device can manage it [6]. With the recent developments in cloud technology since 2010, the gaming industry has started to transition to the cloud gaming business model. According to Cheong et al. [7], cloud gaming is becoming increasingly popular as young people become more accustomed to using technology. There are several benefits of using cloud gaming. Since cloud gaming relies on potent data centers that consolidate gaming hardware, it is more energy-efficient than conventional gaming installations. By centralizing the computing capacity, cloud gaming reduces individual gaming devices' energy consumption, decreasing carbon emissions [8]. Cloud gaming lets users stream games on various devices, including smartphones and old desktops. This extends the hardware life by allowing users to play the most recent games without requiring continual enhancements [9].

Consequently, e-waste production declines, reducing the environmental impact associated with the production and disposal of gaming hardware [10]. With cloud gaming, the required processing capacity to perform games is moved to data centers. This method optimizes the resource allocation because data centers can efficiently distribute the computing power to numerous users. It reduces the need for redundant gaming devices per-device basis, resulting in resource conservation [11]. Overall, cloud gaming offers a promising path to sustainability in the gaming industry, integrating energy efficiency, resource optimization, decreased e-waste, and increased accessibility [12].

Cloud gaming has the potential to transform the gaming industry [13]. Studying acceptance would help assess market demand and adoption rates, allowing companies to make informed decisions about investing in and expanding cloud gaming services [14]. By understanding user acceptance, gaming companies can develop effective strategies to attract and retain customers, potentially leading to business success in this emerging market. Understanding user acceptance would help identify factors that influence the overall experience of cloud gaming. This enables researchers and developers to address the potential barriers and optimize the user interface, game streaming quality, latency issues, and other aspects that impact user satisfaction [15]. By studying acceptance, developers can tailor the gaming experience to meet user expectations and preferences, ultimately improving user engagement and enjoyment.

Many different methods have been used to assess behavioral patterns for technology adoption. The technology acceptance model (TAM) is one notable model from earlier methodologies that attracted much attention due to its practicality and uses. TAM is a model for examining the theory of reasoned action (TRA)-based information system acceptance. In recent decades, TAM has been successfully expanded in numerous study areas. It has provided pertinent applications, illustrating its potential significance in comprehending user behavior towards adopting information technology solutions. The actual system utilization can be predicted by encouraging creative thinking and examining specific impacts on the external variable, such as system features, capacities, etc., early in the TAM model [16]. It is also argued that user encouragement comprises three characteristics that explain actual system usage: perceived ease of use, perceived utility, and attitude. This model utilized attitude to assess the perceived utility and usability, the most reliable indicator of whether a user will accept or reject a system. In addition, several external influences influence perceived utility and simplicity of use. According to Davis [16], perceived usefulness and convenience are the most important factors that each user must consider when deciding whether to support the system. Though the TAM model was the first to be proposed, it has gradually undergone modifications by adding numerous other parameters, such as behavioral intention, which is added to the TAM's fundamental model [16]. Since then, it has been a critical concept in the comprehension of IT system adoption, and it is also thought to be the best model for analyzing the appeal of online games [17].

Researchers claim that several information technology systems use the standard TAM and its extended version. The TAM model has many criteria, such as usability and ease of

use, which directly impact how adaptable the new information system is. Other factors besides the fundamental ones provide the framework for online games that can assess player popularity. Several researchers have already studied and analyzed online gaming and social media. For example, Shin and Shin [18] concentrated on variables that examine a user's intentions when playing online games. To determine why people are interested in using social networks, Lin et al. [19] proposed a model based on motivation theory. While researchers have previously investigated cloud computing and cloud gaming, there is a current literature gap in understanding the aspects that may predict the use of cloud gaming systems. Thus, understanding online gamers' motivations and behavioral intentions to use and accept cloud gaming should be explored.

This study examines the acceptance of cloud gaming in the Philippines and the factors that might affect its adoption among Gen Z gamers using an extended TAM model. Specifically, this study aims to investigate the influence of perceived ease of use, perceived usefulness, perceived trust, and perceived enjoyment on the attitude of Gen Z in using cloud gaming using the structural equation modeling approach. Studying the acceptance of cloud gaming makes several research contributions that advance the understanding of user behavior and the adoption of new technologies. Firstly, examining the acceptance of cloud gaming contributes to the broader field of technology acceptance. By investigating the factors that influence users' intentions to adopt cloud gaming platforms, researchers can deepen their understanding of the determinants of technology acceptance and adoption in general. Secondly, cloud gaming represents a relatively new and evolving technology. Studying its acceptance provides insights into the adoption patterns, challenges, and opportunities associated with this specific domain. Lastly, understanding the user acceptance of cloud gaming would help identify the aspects that contribute to a positive user experience. By investigating the factors such as perceived usefulness, ease of use, and enjoyment, researchers can identify areas for improvement in cloud gaming platforms.

As the gaming current method heavily relies on the device one uses, this study can also determine whether it is a suitable replacement to the traditional method of rendering the game on the device. Cloud gaming is akin to carpooling since multiple users utilize the same servers the company provides. It could reduce e-waste as one server can service many users at any location instead of only 1–3 users being serviced by one PC locally [20].

Research regarding the acceptance of cloud gaming among Gen Z will provide new insights into the general idea of the populace shortly. Through this research, companies offering cloud gaming services would realize the impact of cloud gaming on Filipino youth. They would drive the decision to bring the market to the Philippines and take advantage of how new it is to the people. Economically challenged users would also benefit from this as an alternative before making a significant investment into a better device.

Service providers such as streaming services that could coordinate with or be from gaming companies could take advantage of a new market such as that of the Philippines, where most are limited to handheld devices. The concept of cloud gaming has yet to be introduced to most of the population, yet it could significantly boost sales to place more importance on Southeast Asia. Finally, this research study could also serve as a reference and data point for researchers in cloud gaming regarding Filipino youth, serving as a guide for other researchers with the variables being used.

Figure 1 shows the proposed conceptual framework for the study using the technology acceptance model (TAM). TAM is a reliable model for examining the acceptability of new technologies and defining user behavior and technology usage [21,22]. Because TAM was frequently investigated in the context of other technologies unrelated to cloud gaming applications, the present study intends to explore other constructs related to user acceptability by extending the fundamental TAM model.

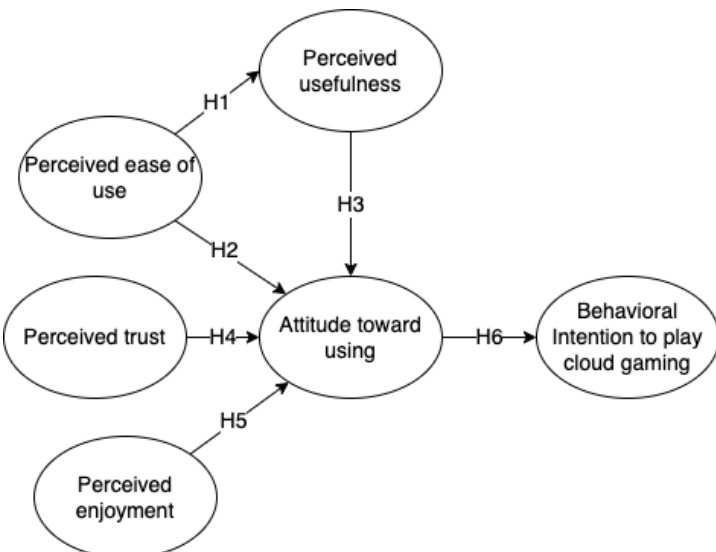

**Figure 1.** Proposed conceptual framework based on the technology acceptance model (TAM).

The technological acceptance model (TAM) has been used in several earlier studies to examine the adoption of mobile games [23]. The extended TAM was utilized in this study's analysis of cloud gaming acceptance. Perceived utility and perceived usability are fundamental elements in technology adoption, according to TAM theory. These elements may impact the attitude that influences the playing intention [24]. The expanded TAM has not previously been employed in prior studies that include the two crucial aspects in the acceptance of cloud gaming: perceived trust and enjoyment.

The research model being proposed is an extension of the traditional TAM paradigm. The belief-attitude-intention-behavior causal chain hypothesis is therefore applied to the context of cloud gaming. In this present study, the perceived ease of use (PEU) is used to measure how much work a user perceives is required to play a game because gaming services are designed to be entertaining. A high PEU rating suggests that the game's rules are simple to learn and follow. Since mobile online games are entertainment-based services, ease of use will help to show how often a user can access an online game. A high perceived ease of use (PEU) score indicates that playing an online game and understanding the rules are both simple processes.

On the other hand, perceived usefulness (PU) and perceived enjoyment (PE) are also critical components of TAM. Perceived usefulness (PU) is defined as the player's perception of a better quality of life due to cloud gaming. At the same time, an action thought to be pleasurable and enjoyable without consequences is known as perceived enjoyment (PE). PE is proven to significantly and positively impact players' attitudes and willingness to follow an online game's rules [25]. Thus, PE is a characteristic of gaming and perceptual enjoyment that should be considered in gaming research and practice [26].

Trust is also a significant factor in the consumer acceptance of technology [27]. Players' gaming experiences and interests can be significantly hampered by network congestion, server overload, or server failure [28]. As a result, the users of online games could wonder whether remote servers can deliver high-quality online gaming services. In the context of online gaming, these issues and concerns necessitate the engagement of trust.

Thus, combining the TAM model of the existing research results, it was hypothesized that:

**H1.** *Perceived ease of use (PEU) has a significant and positive effect on perceived usefulness (PU).*

**H2.** *Perceived ease of use (PEU) has a significant and positive effect on attitude (AT) towards using cloud gaming.*

**H3.** *Perceived usefulness (PU) has a significant and positive effect on attitude (AT) towards using cloud gaming.*

**H4.** *Perceived trust (PT) has a significant and positive effect on attitude (AT) towards using cloud gaming.*

**H5.** *Perceived enjoyment (PE) has a significant and positive effect on attitude (AT) towards using cloud gaming.*

**H6.** *Attitude (AT) towards using cloud gaming has significant and positive effect on behavioral intention (BI) to play cloud gaming.*

This study follows a standard structure with several sections to effectively present the research. The article begins with a concise and informative introduction section that accurately reflects the content of the study. Towards the end of the introduction, the objectives and hypothesis of the study are stated to outline the purpose and direction of the research.

The materials and methods section to describe the study design, methodology, data collection procedures, and statistical analyses is presented in the next section. The study findings are presented in a clear and concise manner. The discussion section follows the results and involves interpreting and analysis of the results. Finally, the conclusion section summarizes the study's main findings and reiterates their significance.

## 2. Materials and Methods

### 2.1. Participants

Due to the unknown population of players of the cloud gaming, a convenience sampling technique was employed for data collection in this study. The target respondents are Gen Z users in the NCR region. The NCR region had the greatest GDP in 2016 and it also had the lowest poverty incidence, at 3.9 percent, compared to the national average of 21.6 percent and just a tenth of the average of 37.1 percent for Mindanao (where the Autonomous Region in Muslim Mindanao is worst, at 53.7 percent). Since the NCR is the biggest among the other regions, it is far more urbanized. In today's cities, technology is pervasive and has long been applied to solve certain urban problems. Cities are ideal places to adopt new technologies due to their high population density and infrastructure density, which results in lower implementation costs per capita [29]. The expected minimum number of respondents are 300, as suggested by the study of Yamane [30], where the level margin of error was set at 10%. In general, the sample size's square root determines how precise an estimate will be, therefore, to double the precision, the sample size was quadrupled. In general, sample sizes of 200–300 participants offer a respectable margin of error and are below the declining returns threshold.

### 2.2. Data Gathering Tools

The online survey was conducted during Q1 of 2022 through the use of self-administered type and distributed via a Google form. The questionnaire was distributed with multiple cross-sectional designs and the survey was sent to the target respondents for two months. The questionnaire is presented in the English language.

The survey consisted of 36-item questions. The respondent's demographics were determined in the first section of the questionnaire using 6-item questions, including age, gender, educational level, residential area, hours spent in online gaming, and years of playing online games.

The second part of the questionnaire consists of the indicators based on the extended TAM model: perceived ease of use, perceived usefulness, perceived trust, perceived enjoyment, and attitude towards using cloud gaming. This measured the users' perceived behavioral intention to play cloud gaming. The survey consists of item questions where all answers were on a 5-point Likert scale ranging from "strongly disagree" to "strongly

agree". Six (6) latent were used in the survey. The summary of measures and constructs is shown in Table 1. The items for the constructs were adopted from existing studies.

**Table 1.** Summary of Constructs and Measurement Items.

| Items | Measure | Supporting References |
|---|---|---|
| | Perceived Ease of Use | |
| PEOU1 | I find it easy to learn how to play in the cloud game. | |
| PEOU2 | I find it easy to be good at playing in the cloud game. | |
| PEOU3 | I find it easy to do what I want to do in the cloud game. | [31–34] |
| PEOU4 | I find it easy to play in the cloud game. | |
| PEOU5 | Interactions in the cloud game are clear and easy to understand. | |
| | Perceived Usefulness | |
| PU1 | I find it easy to remember the steps on how to play in the cloud game. | |
| PU2 | Playing in the cloud game is one way of getting new friends for me. | |
| PU3 | Playing in the cloud game can improve my imagination skills. | [31–34] |
| PU4 | By playing in the cloud game, I find it easier to communicate with my friends. | |
| PU5 | Playing in the cloud game may improve my skills, for example, my English language skills. | |
| | Perceived Trust | |
| PT1 | The developer of cloud game is trustworthy. | |
| PT2 | I believe in the information that the cloud game provides. | |
| PT3 | The cloud game has adequate security features. | [32–37] |
| PT4 | I feel like my privacy is protected in the cloud game. | |
| PT5 | I feel I can trust the cloud game. | |
| | Perceived enjoyment | |
| PE1 | It would be fun to play cloud games. | |
| PE2 | I will not be bored while playing cloud games. | |
| PE3 | Cloud gaming will make my leisure time more fun. | [38,39] |
| PE4 | I am interested in playing cloud games. | |
| PE5 | I enjoy playing cloud games. | |
| | Attitude | |
| AT1 | Playing cloud games is good. | |
| AT2 | Playing cloud games is pleasant. | |
| AT3 | Playing cloud games is interesting. | [37,40,41] |
| AT4 | It is a good idea for me to play cloud games during my free time. | |
| AT5 | I feel good towards cloud games. | |
| | Behavioral intention | |
| BI1 | I intend to play cloud game often. | |
| BI2 | I am likely to play cloud game in the near future. | |
| BI3 | I wish that my cloud game play habit continues in the future. | [31,42,43] |
| BI4 | I am motivated to use cloud game. | |
| BI5 | I recommend using cloud games. | |

### 2.3. Data Analysis

Multivariate analysis assessed survey data. PLS-SEM with maximum likelihood estimation was employed in this investigation. PLS-SEM analyzes abstract concept relationships [44]. It predicts complex constructs with greater degrees of abstraction and yields better construct reliability and validity, making it beneficial in our investigation [45]. Its fundamental purpose is to explain the dependent construct variation as much as feasible. Measurement model characteristics also determine data quality. PLS-SEM evaluates both direct and indirect effects on presumed causal linkages, according to Ouellette and Wood [46]. PLS-SEM is ideal for creating new theories and producing predictions with a factor loading of 0.7 or higher, whereas CB-SEM is better for testing and verifying existing

hypotheses [44]. PLS-SEM was used to justify model fit using SRMR, NFI, and chi-square. SRMS should be less than 0.08 [47]. Baumgartner and Homburg [48] recommended an NFI value of 0.80 or above and a chi-square value below 5.0.

## 3. Results

### 3.1. Profile of the Respondents

A total of 356 respondents answered the survey, 70.3% being male, 28.4% being female, and 1.2% identifying as genderfluid or they/them. As for the highest educational level attained, 58.7% of the respondents were currently in senior high school while 28.4 were in college. Among the respondents, 9.7% were at junior high school while only 3.2% of the respondents had graduated. Moreover, 40% of the respondents gamed for 2–4 h per day while 27.1% gamed less at 1–2 h per day. Additionally, 17.4% played games for less than an hour while 10.3% played games for 4–8 h or more daily. Only 5.2% of the respondents played games daily for 9 h or more. Most of the respondents at 52.9% used cloud gaming before while 47.1% have used cloud gaming previously.

Based on the profiles of the respondents, the ratio between males to females is significantly higher on the male side compared to the female side. According to Duggan [49], 50% of men and 48% of women play video games. This result does not align with the previously stated research as the respondents at that time in that area might have been more male-dominated. Furthermore, the respondents show that there is no correlation between the genders and their knowledge of cloud gaming.

### 3.2. Validity of the Results

The graphical representation of a model in determining the factors affecting the behavioral intentions of Gen Z to use cloud gaming is presented in Figure 2. The model has six latent variables and 30 indicators. Table 2 shows the model's factor loading, reliability, and validity. Reliability analysis must precede structural equation modeling (SEM). Cronbach's alpha, CR, and AVE are used to analyze behavioral intentions models. Cronbach's $\alpha$, CR, and AVE must exceed 0.7 [44,50]. This model's constructs are valid and dependable since all values exceed requirements.

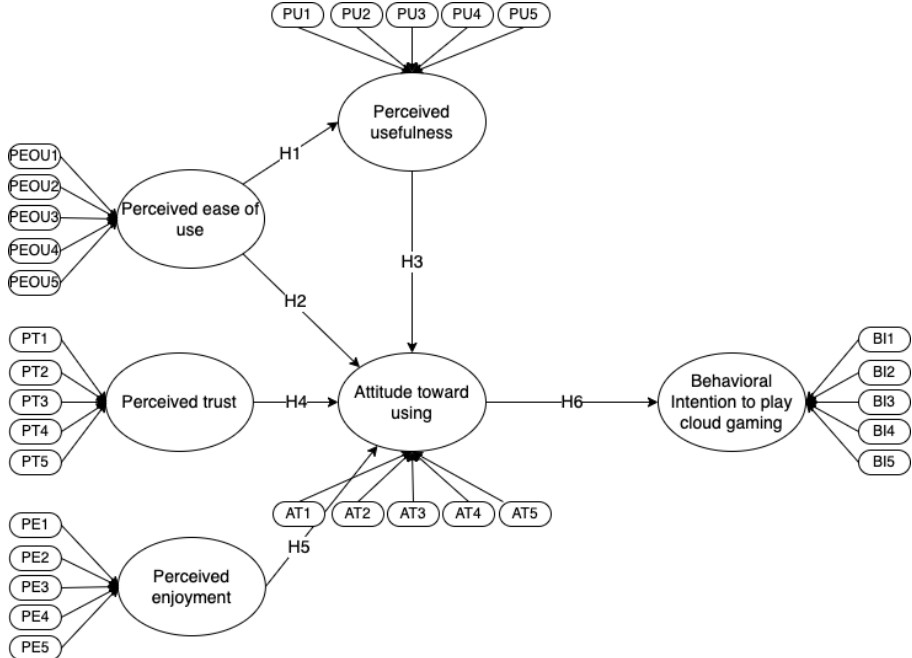

**Figure 2.** The initial SEM for determining the factors affecting the intention to use cloud gaming.

**Table 2.** Reliability and convergent validity result.

| Construct | Items | Mean | S.D. | FL ($\geq$0.7) | $\alpha$ ($\geq$0.7) | CR ($\geq$0.7) | S.D. |
|---|---|---|---|---|---|---|---|
| Perceived Ease of Use (PEOU) | PEOU1 | 3.51 | 1.03 | 0.812 | | | |
| | PEOU2 | 3.43 | 1.07 | 0.804 | | | |
| | PEOU3 | 3.21 | 1.07 | 0.856 | 0.875 | 0.882 | 0.666 |
| | PEOU4 | 3.60 | 0.99 | 0.844 | | | |
| | PEOU5 | 3.56 | 1.03 | 0.764 | | | |
| Perceived Usefulness (PU) | PU1 | 3.68 | 1.04 | 0.757 | | | |
| | PU2 | 3.54 | 1.09 | 0.829 | | | |
| | PU3 | 3.56 | 1.11 | 0.885 | 0.896 | 0.821 | 0.757 |
| | PU4 | 3.85 | 1.11 | 0.882 | | | |
| | PU5 | 3.57 | 1.02 | 0.846 | | | |
| Perceived Trust (PT) | PT1 | 3.47 | 1.00 | 0.870 | | | |
| | PT2 | 3.37 | 0.96 | 0.858 | | | |
| | PT3 | 3.46 | 0.94 | 0.891 | 0.920 | 0.921 | 0.757 |
| | PT4 | 3.40 | 1.07 | 0.866 | | | |
| | PT5 | 3.56 | 1.06 | 0.864 | | | |
| Perceived Enjoyment (PE) | PE1 | 3.60 | 1.07 | 0.875 | | | |
| | PE2 | 3.63 | 1.06 | 0.842 | | | |
| | PE3 | 3.62 | 1.03 | 0.890 | 0.924 | 0.943 | 0.767 |
| | PE4 | 3.65 | 1.05 | 0.836 | | | |
| | PE5 | 3.57 | 1.06 | 0.934 | | | |
| Attitude (AT) | AT1 | 3.30 | 1.06 | 0.804 | | | |
| | AT2 | 3.33 | 1.05 | 0.832 | | | |
| | AT3 | 3.64 | 1.07 | 0.829 | 0.899 | 0.904 | 0.714 |
| | AT4 | 3.67 | 0.97 | 0.845 | | | |
| | AT5 | 3.59 | 1.05 | 0.911 | | | |
| Behavioral Intention (BI) | BI1 | 3.41 | 1.11 | 0.920 | | | |
| | BI2 | 3.30 | 1.11 | 0.893 | | | |
| | BI3 | 3.51 | 1.11 | 0.891 | 0.932 | 0.934 | 0.787 |
| | BI4 | 3.45 | 1.06 | 0.874 | | | |
| | BI5 | 3.61 | 1.03 | 0.858 | | | |

### 3.3. Hypothesis Test

The PLS-SEM was performed to test the proposed hypotheses using Smart PLS v3.3.3. The results are shown in Table 3. It could be seen that behavioral intention to use cloud gaming among Filipino youth was significantly influenced by attitude ($\beta$ = 0.877, $p$ < 0.001), while attitude was significantly influenced by perceived enjoyment ($\beta$ = 0.660, $p$ < 0.001) and perceived usefulness ($\beta$ = 0.253, $p$ = 0.013). Moreover, it was also found that the perceived usefulness was influenced by perceived ease of use ($\beta$ = 0.731, $p$ < 0.001). On the contrary, perceived trust ($\beta$ = 0.048, $p$ = 0.494) and perceived ease of use ($\beta$ = 0.091, $p$ = 0.183) are found to have a no significant influence on behavioral intentions to use loud gaming.

**Table 3.** Hypothesis Test.

| No. | Relationship | Beta Coefficient | *p*-Value | Result | Significance | Hypothesis |
|---|---|---|---|---|---|---|
| 1 | PEOU→PU | 0.731 | <0.001 | Positive | Significant | Accept |
| 2 | PEOU→AT | 0.091 | 0.183 | Positive | Not significant | Reject |
| 3 | PU→AT | 0.253 | 0.013 | Positive | Significant | Accept |
| 4 | PT→AT | 0.048 | 0.494 | Positive | Not significant | Reject |
| 5 | PE→AT | 0.660 | <0.001 | Positive | Significant | Accept |
| 6 | AT→BI | 0.877 | <0.001 | Positive | Significant | Accept |

Table 3 shows that four components positively affected BI. Two constructs did not affect behavioral intentions (BI). The model is robust since four of the six hypotheses substantially affected behavioral intentions and two did not [51]. Attitude (AT) influences behavioral intentions (BIs) most directly, followed by perceived pleasure (PE) and perceived usefulness (PU).

Discriminant validity utilizing the Fornell–Lacker criteria and the heterotrait–monotrait ratio of correlation are used to verify substantial correlation between the factors and assess the measurement model [52]. The values are within the intended range, and Tables 4 and 5 show good reliability and convergent validity. Thus, all build findings are acceptable.

**Table 4.** Discriminant Validity: Fornell–Larcker Criterion.

|  | **AT** | **BI** | **PEOU** | **PE** | **PU** | **PT** |  |
|---|---|---|---|---|---|---|---|
| AT | 0.845 |  |  |  |  |  | AT |
| BI | 0.816 | 0.887 |  |  |  |  | BI |
| PEOU | 0.729 | 0.724 | 0.816 |  |  |  | PEOU |
| PE | 0.809 | 0.862 | 0.740 | 0.876 |  |  | PE |
| PU | 0.747 | 0.817 | 0.731 | 0.834 | 0.841 |  | PU |
| PT | 0.700 | 0.701 | 0.717 | 0.756 | 0.727 | 0.870 | PT |

**Table 5.** Discriminant Validity: Heterotrait–Monotrait Ratio.

|  | **AT** | **BI** | **PEOU** | **PE** | **PU** | **PT** |  |
|---|---|---|---|---|---|---|---|
| AT |  |  |  |  |  |  | AT |
| BI | 0.756 |  |  |  |  |  | BI |
| PEOU | 0.818 | 0.801 |  |  |  |  | PEOU |
| PE | 0.793 | 0.727 | 0.818 |  |  |  | PE |
| PU | 0.743 | 0.815 | 0.818 | 0.738 |  |  | PU |
| PT | 0.766 | 0.756 | 0.797 | 0.819 | 0.800 |  | PT |

Figure 3 shows the completed SEM model. Beta coefficients and R2 values assessed the hypothesis model. The model assigns 38.7% to intention to utilize cloud gaming, 74.2% to behavioral intention, 86.6% to attitude, and 53.4% to perceived usefulness. The variation in the model is deemed high and acceptable following the study of Hair et al. [44].

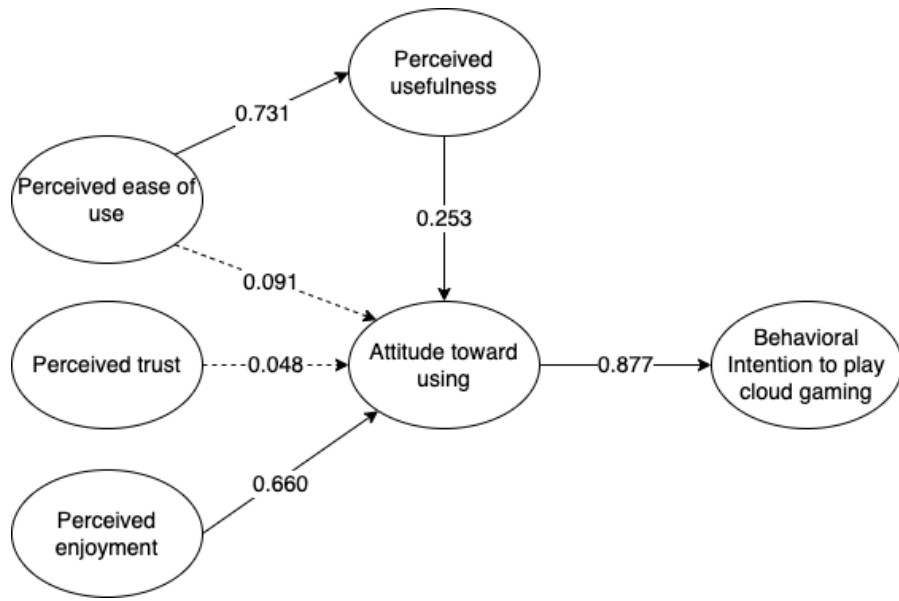

**Figure 3.** The final SEM for determining the factors affecting the intention to use cloud gaming.

Model fit analysis validated the proposed model. Using prior model fit parameters, this research used SRMR, chi-square, and NFI [47,48]. Table 6 shows that all parameter estimations surpassed the minimal criterion, validating the suggested model.

**Table 6.** Model Fit.

| Model Fit for SEM | Parameter Estimates | Minimum Cut-Off | Recommended by |
|---|---|---|---|
| SRMR | 0.062 | <0.08 | [47] |
| (Adjusted) chi-square/dF | 4.03 | <5.0 | [47] |
| Normal fit index (NFI) | 0.921 | >0.90 | [48] |

## 4. Discussion

Internet gaming is a rapidly expanding industry in the Philippines, with revenue expected to increase in the following years compared to the past. Gaming on consoles, mobile devices, and personal computers is gaining popularity among Filipinos, especially the younger generation. As the cost of GPUs and other computer components continues to climb, cloud gaming is considered the future of the industry. The notion of cloud gaming has been around for several years; however, there currently needs to be an official cloud gaming service in the Philippines. With this, the present study aims to determine the factors that affect the intention of Filipino youth to use cloud gaming by using the extended technology acceptance model (TAM). Structural equation modeling (SEM) was utilized to determine the factors affecting Filipino youths' behavioral intentions toward cloud gaming. Numerous latent factors were used in the analysis, such as perceived ease of use (PEOU), perceived trust (PT), perceived enjoyment (PE), perceived usefulness (PU), attitude (AT), and behavioral intention (BI) for cloud gaming.

From the results, it could be seen that attitude (AT) has the highest significant and direct influence on the behavioral intention (BI) to use cloud gaming, thereby accepting H1. According to prior studies, attitude toward technology is described as the entire affective response of users, especially youth, to accessing cloud services [23,53]. The favorable or negative attitudes of users toward cloud services may influence their behavioral intentions. In another study, Arpaci et al. [54], found that attitude significantly determines cloud gaming adoption. According to Kotchen and Reiling [55], attitude is a significant predictor of an action's intent. Attitude is a psychological emotion that is transmitted by the consumers' evaluations. If it is favorable, it tends to positively affect the intent to act [56]. In particular, there is a positive correlation between attitude and behavior [57]. Moreover, Van Birgelen et al. [58] stated that a user's positive opinion toward a product would enhance their intent to use it. Thus, for users to have a higher intention to utilize cloud gaming, it is necessary to examine their beliefs and attitudes regarding cloud gaming and determine their demand for such a system to optimally implement it. Users must be well versed in the system and understand how to optimize its utility. The system should not be complicated and simple to use. The findings of the study contribute to new knowledge in the field of technology acceptance. Understanding the relationship between attitude and intention to use cloud gaming can provide valuable insights into user behavior and inform the development and adoption of cloud gaming platforms. By examining the specific context of cloud gaming, researchers can further validate and refine TAM and related theories, enhancing the understanding of the underlying psychological factors that drive technology acceptance.

Perceived enjoyment (PE) was also proved to have a significant positive influence on behavioral intention (BI) to use cloud gaming, thereby accepting H4. This suggests that the satisfaction aspects associated with cloud gaming play a major role in influencing the behavioral intention to embrace cloud games. Several studies conducted in both developed and developing countries have demonstrated the impact of perceived enjoyment in positively influencing behavioral intention to adopt cloud gaming. In Jordan, Faqih [59] explored the adoption of mobile AR games from a developing country perspective. The

analysis revealed that perceived enjoyment is one of the factors that affect user's behavioral intention to adopt mobile AR games. This implies that individuals are more likely to accept new mobile-based technologies if they find them enjoyable, gratifying, and pleasant. It has been acknowledged that the aspects of enjoyment play a vital influence in boosting the acceptance and implementation of technological innovation, particularly gaming technologies [60]. As such, in order to optimize players' levels of enjoyment, engagement, and entertainment, game creators should create unique ways to reinforce distinct patterns and components of enjoyment throughout gaming situations. The impact of perceived enjoyment on the intention to use cloud gaming are important aspects of research in the field of technology acceptance and contributes to existing knowledge. Understanding the effect of perceived enjoyment could help guide the design and development of user-centered features, content, and gaming experiences that improve user involvement and drive growth.

Perceived usefulness (PU) was also found to have a significant positive effect on behavioral intention (BI) to use cloud gaming, thereby accepting H5. Prior studies have determined that perceived usefulness is the most significant predictor of behavioral intention to accept and utilize a new technology [61–63]. Because of its overwhelming influence on behavioral intention, the inclusion of the perceived utility parameter is an absolute prerequisite for any investigation into the adoption of a new technology. In fact, empirical studies undertaken across I.T. disciplines and in many cultural contexts have established that the perceived usefulness and behavioral purpose are most likely positively connected [59,64,65]. This suggests that, if the perceptions of usefulness are strengthened, the intention to embrace the technology will increase, resulting in an increase in the rate of adoption of new technology. The observed findings are consistent with prior studies undertaken in developing country settings [66,67]. Thus, strategies must be developed to improve the perceptions of usefulness. This can be accomplished by gamers recognizing that the technology is very beneficial, technically aligned with their gaming performance, and interacting with their experience in a seamless manner. In addition, the e-technologies responsible for implementing cloud gaming systems, such as websites and their associated tools, must be designed to promote user experience as well as be highly usable, effective, and efficient environments, and an innovative web interface design that maximizes flexibility leverage. Understanding the influence of perceived usefulness on the intention to use cloud gaming provides valuable insights into user behavior and sheds light on the factors that drive the adoption and usage of cloud gaming platforms. By incorporating features and capabilities that are perceived as useful, platforms can enhance user experience and satisfaction, leading to increased adoption and long-term usage.

In addition, perceived ease of use (PEOU) was proved to have a significant direct effect on perceived usefulness (PU), thereby accepting H3. This result is similar to previous studies that demonstrated the association between PEOU and PU [16,68,69]. This suggests that, if users perceive cloud gaming to be user-friendly, they will be more inclined to utilize it for gaming purposes. TAM proposes that both PEOU and PU impact the establishment of favorable attitude linked with the use of technology, which, when combined with PU, increases persons' BI to utilize the technology. In addition, PEOU is anticipated to positively influence users' perceptions of the technology's usage intention [53]. According to a study by Gefen and Straub [70], the perceived ease of use has a substantial impact on e-commerce when the reason for visiting the website is to seek information, whereas perceived usefulness has a greater role when the reason for visiting the website is to make a purchase. So, when applying TAM to situations where new technologies are deployed, perceived ease of use and perceived usefulness encourage the adoption of new technologies such as cloud gaming. This indicates that perceived ease of use increases the influence of perceived usefulness on technology adoption. Perceived ease of use captures users' subjective perceptions of the ease and simplicity of interacting with cloud gaming platforms. By studying the impact of perceived ease of use, researchers can gain insights into the usability factors that influence users' perceptions of the ease of learning and using

cloud gaming platforms. The findings of the study could provide a valuable contribution to a new knowledge in technology acceptance.

On the contrary, perceived trust (PT) and perceived ease of use (PEOU) were found to have no significant influence on the attitude (AT) towards the use of the cloud gaming, thereby rejecting H2 and H6. These findings contradict the result of prior studies that proved the association between perceived trust and ease of use to attitude. This shows that the perception of users to take security risks in using cloud gaming, as well as the degree to which the user believes that using gaming will be an effort-free, have no direct effect on the attitude of users towards cloud gaming. One reason we did not find support for the role of perceived trust and perceived ease of use may be because cloud gaming is not yet popular in the country and has only been adopted by a small number of people. While perceived trust is typically recognized as a crucial factor in technology acceptance and adoption, the lack of significant impact in the context of cloud gaming presents opportunities for further exploration and improvement, especially to the gaming industry. The gaming industry can use this insight to focus on enhancing other aspects of the user experience that have a more significant impact on users' attitudes. This may entail enhancing the overall quality of games, providing a variety of gaming experiences, ensuring seamless gameplay, or ensuring high-performance broadcasting and low latency.

Consequently, perceived ease of use and trust may not have been a significant issue for the Filipino user, even if they had used the product previously. As such, the Philippines should focus on enhancing and modernizing its network infrastructure to increase the overall speeds and latency to accommodate cloud gaming. In terms of latency, satellite internet services, such as Starlink, should improve less than land-based infrastructures. The insignificant relationship between perceived ease of use and attitude suggests that users are either already familiar with cloud gaming or are willing to invest the time and effort required to learn and acclimate to new platforms. Therefore, user education and communication efforts can emphasize the unique benefits and advantages of cloud gaming instead of merely focusing on simplicity of use. The gaming industry should educate users on the convenience, accessibility, cost-effectiveness, and game library options offered by cloud gaming platforms to help shape favorable attitudes and increase adoption.

Hence, the findings of this study contribute to the existing body of knowledge regarding the influence of attitude, perceived enjoyment, perceived usability, perceived ease of use, and perceived trust on the acceptance and adoption behavior of cloud gaming.

## 5. Conclusions

This study is one of the first to determine the acceptance of cloud gaming and its factors that would lead its greater use by Gen Z in the Philippines. Using the PLS-SEM, it was found that the perceived trust (PT) and the perceived ease of use (PEOU) did not have a notable effect on the attitude towards using cloud gaming. However, the perceived ease of use (PEOU) positively affects the perceived usefulness (PU) and that the PU and the perceived enjoyment (PE) both have a significantly positive effect on the attitude towards using cloud gaming. In conclusion, the survey conducted on the use of cloud gaming has provided valuable insights into the gaming habits and preferences of the respondents. The results show that the respondents were male and were currently in senior high school. This demographic composition may suggest that the survey mainly targeted a younger audience and may not represent the gaming population.

Additionally, the survey indicates that a significant proportion of the respondents played games for 2–4 h per day, while a smaller percentage of respondents played for longer periods of time. This suggests that most respondents viewed gaming as a form of leisure and not a major time commitment. The fact that more than half of the respondents have used cloud gaming before also shows a growing interest in this technology among gamers.

One interesting finding is that there was no significant difference in the knowledge of cloud gaming between male and female respondents. This contradicts previous research

that suggested that gaming was predominantly male-dominated and suggests that cloud gaming is an area with a level playing field between the genders.

Despite the interesting insights gained from the study, it is important to note that the results may not be entirely representative of the wider population of gamers. The survey was conducted in a specific geographical location and targeted a specific age group, which may limit the generalizability of the findings. Future research should aim to conduct similar surveys across different demographics and locations to provide a more comprehensive understanding of the use of cloud gaming.

*5.1. Practical Implications*

Cloud gaming is a rapidly growing industry that is changing the way people play and consume video games. The results of this study would help cloud service providers, game publishers, and security experts learn more about the factors that predict how much people enjoy online gaming. The study's results could also help companies decide whether to move their low-latency apps to a cloud subscription service model. The practical implications of cloud gaming studies suggest that the technology has the potential to revolutionize the gaming industry by making high-quality games more accessible and affordable to a broader audience. Should the demand for better cloud gaming services in the Philippines increase, companies such as Microsoft, Google, Sony, and Geforce, who are in that field, could take advantage of how the technology is relatively new in the Philippines. As such, the companies could use the result of this study to meet those qualifications to attract the population.

*5.2. Theoretical Implications*

The study's findings have significant implication for cloud gaming platforms and highlight the necessity for future investigations into cloud gaming evaluation techniques. Because developers and cloud gaming providers must anticipate user behavior in order to obtain a competitive edge, research on cloud gaming acceptance predictors is vital. This study utilized the technology acceptance model to identify the elements that influence the behavioral intentions of Filipino youth to use cloud gaming. Due to the paucity of research in the Philippines, this paper presents a more thorough framework for describing the intention to use cloud gaming than previous research. As a result, the findings of this study can serve as a theoretical framework for future consumer behavior researchers, paving the way for the deployment of cloud gaming technologies in the Philippines as part of its rising economy.

*5.3. Limitations*

Although this study obtained excellent results, a few things must be considered. First, the sampling technique employed in this study is convenience sampling. This was chosen due to its quick and accessible data collection method, allowing the researchers to reach participants who are readily available or easily accessible. However, one of the primary limitations of convenience sampling is the potential for sample bias. Participants who are easily accessible or willing to participate may not represent the larger population. This can introduce a selection bias, leading to results that may not accurately reflect the broader population's views, characteristics, or behaviors. Thus, it is suggested to consider replicating the study using more rigorous and representative sampling methods, such as random, stratified, or cluster sampling. Conducting similar research with a more representative sample can help validate or refine the initial findings and enhance generalizability. Second, the survey focuses on respondents who are members of Generation Z (24 years old); therefore, it is recommended to distribute the survey to older generations (24 years and older), as they are also potential cloud gaming users. Through an online survey, this study considered a quantitative analysis of the technology acceptability model. Future researchers may use interviews to combine quantitative and qualitative findings. Using this technique, additional factors may be uncovered. Additionally, future research should

include other variables, such as an individual's cloud gaming experience, as this may also impact their overall acceptability. Since most respondents resided in urban areas, particularly NCR, it is suggested that future research evaluate cloud gaming acceptability in rural and provincial areas. This research examined an expanded TAM framework. Other analysis techniques, such as machine learning algorithms, should be considered to validate the framework developed and utilized in this research.

**Author Contributions:** Conceptualization, M.J.J.G.; Methodology, R.F.B.A., J.M.V.N. and K.L.D.G.; Software, R.F.B.A., J.M.V.N. and K.L.D.G.; Validation, M.J.J.G.; Formal analysis, M.J.J.G.; Resources, J.M.V.N. and K.L.D.G.; Data curation, R.F.B.A.; Writing—original draft, R.F.B.A., J.M.V.N. and K.L.D.G.; Writing—review & editing, M.J.J.G.; Project administration, M.J.J.G.; Funding acquisition, M.J.J.G. All authors have read and agreed to the published version of the manuscript.

**Funding:** This research was funded by Mapua University Directed Research for Innovation and Value Enhancement (DRIVE) grant number (FM-RS-03-25).

**Institutional Review Board Statement:** This study was approved by Mapua University Research Ethics Committee.

**Informed Consent Statement:** Informed consent was obtained from all subjects involved in the study.

**Data Availability Statement:** The data presented in this study ae available on request from the corresponding author.

**Conflicts of Interest:** The authors declare no conflict of interest.

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
