# Peer review of "Antecedents of Cloud Gaming Acceptance among Gen Z: Achieving Sustainability in the Digital Gaming Industry"

_sustainability, doi:10.3390/su15129189_

Round 1

Reviewer 1 Report

Thank you for the opportunity to review this article, entitled: Antecedents of Cloud Gaming Acceptance among Gen Z: Achieving Sustainability in the Digital Gaming Industry. 

The topic is very new and topical. 

The idea of the paper is interesting and with good results about cloud gaming and its sustainability by decreasing e-waste and reducing environmental impact.

The aim of the paper is to determine the acceptance of Gen Z towards cloud gaming as a potential solution to achieve sustainability in the digital games industry. The study uses a causal research design. To do so, it uses the extended technology acceptance model (TAM) by implementing the partial least structural equation model, PLS-SEM (using Smart PLS v3.3.3).

To determine the factors affecting Filipino youth, from the NCR region, behavioural intentions towards cloud gaming. Numerous latent factors are used. The model consists of 6 latent variables and 30 indicators.

Although the sample is small, 356 respondents. It justifies it with Yamane's study (minimum of 300). The authors point out that this is a limitation and could be extended in future work to generalise the results. Also, the survey was conducted in a very specific geographical location, the NCR region of the Philippines, and targeted a specific age group.

The methodology implemented is adequate.

The results of the research will help cloud service providers and game publishers.

Regarding the bibliographic references, a large majority of them are very recent.

Some recommendations for improving the study are given below: 

MAIN CONCERNS: 

1. The research hypotheses, as well as the objectives, should be included in paragraph 1 of the text of the article. 

2. It is also good practice to state the structure of the article in the last paragraph of the introductory section.

Therefore, it would be acceptable for publication as it stands despite the sample size, age of respondents and geographical area. Nevertheless, it may serve as a future theoretical framework for other researchers.

Reviewer 2 Report

Comment 1: The introduction contains several instances of redundant information and repetitive statements. It is essential to streamline the content and remove repetitive sentences to maintain a clear and concise narrative.

Research gap: While the introduction mentions a research gap in understanding the aspects that may predict the use of cloud gaming systems, it does not explicitly define this gap or state how the current study aims to address it. Clarifying the specific research objectives and contributions of the study would enhance the introduction's clarity and relevance.

Wordiness and clarity: Some sentences in the introduction are excessively wordy and could be simplified to improve clarity. It is important to ensure that the language used is concise and accessible to readers.

Comment 2: the Sampling technique and population mention the use of convenience sampling technique for data collection. While convenience sampling can be suitable in certain circumstances, it is important to acknowledge its limitations in terms of generalizability. The unknown population of cloud gaming players and the specific target respondents being Gen Z users in the NCR region raise questions about the representativeness of the sample. It would be beneficial to discuss the potential implications of using convenience sampling and address any limitations in generalizing the findings to a broader population.

Comment 3: in the discussion, It would be valuable to provide a deeper understanding of the implications of the findings and how they contribute to the existing knowledge in the field.

Comment 4: The study's limitations, such as the specific age group and geographical location, should be emphasized more strongly in the paper to avoid potential misinterpretation and generalization of the findings.

Minor editing of English language required

Reviewer 3 Report

Thank you for this interesting research. This study aims to examine the acceptance of cloud gaming in the Philippines and the factors that might affect its adoption among Gen Z gamers. Hereby are some comments that may help you improve on it:

1.      For the hypothesis H2 and H4, the results are reject, please add more discussion for them. According these reject results, how about the contributions from the gaming industry perspective?

2.      Regarding the practical implications and practical implications, I suggest the authors could strength them. In addition, the heading number of the 4.1 Practical Implications and 4.2 Theoretical Implications are wrong, please check them and revised.

I hope that these notes are helpful in reviewing your article.
